# PsyCoT: Psychological Questionnaire as Powerful Chain-of-Thought for Personality Detection

**Tao Yang[1], Tianyuan Shi[1], Fanqi Wan[1], Xiaojun Quan[1]***
**Qifan Wang[2], Bingzhe Wu[3], Jiaxiang Wu[3]**

[1]School of Computer Science and Engineering, Sun Yat-sen University, China
[2]Meta AI, [3]Tencent AI Lab

{yangt225, wanfq, shity6}@mail2.sysu.edu.cn, quanxj3@mail.sysu.edu.cn
wqfcr@fb.com, wubingzheagent@gmail.com, jonathanwu@tencent.com

## Abstract

Recent advances in large language models (LLMs), such as ChatGPT, have showcased remarkable zero-shot performance across various NLP tasks. However, the potential of LLMs in personality detection, which involves identifying an individual's personality from their written texts, remains largely unexplored. Drawing inspiration from Psychological Questionnaires, which are carefully designed by psychologists to evaluate individual personality traits through a series of targeted items, we argue that these items can be regarded as a collection of well-structured chain-of-thought (CoT) processes. By incorporating these processes, LLMs can enhance their capabilities to make more reasonable inferences on personality from textual input. In light of this, we propose a novel personality detection method, called **PsyCoT**, which mimics the way individuals complete psychological questionnaires in a multi-turn dialogue manner. In particular, we employ a LLM as an AI assistant with a specialization in text analysis. We prompt the assistant to rate individual items at each turn and leverage the historical rating results to derive a conclusive personality preference. Our experiments demonstrate that PsyCoT significantly improves the performance and robustness of GPT-3.5 in personality detection, achieving an average F1 score improvement of 4.23/10.63 points on two benchmark datasets compared to the standard prompting method. Our code is available at https://github.com/TaoYang225/PsyCoT.

## 1 Introduction

Personality, as an important psychological construct, refers to individual differences in patterns of thinking, feeling, and behaving (Corr and Matthews, 2009). Consequently, detecting one's personality from their generated textual data has garnered considerable interest from researchers due to its wide-ranging applications (Khan et al., 2005;

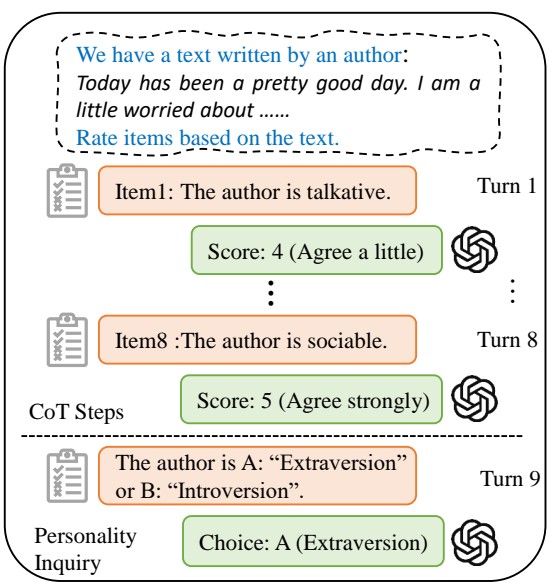

Figure 1: An illustration of our PsyCoT, where items (1-8) from the personality questionnaire are employed as the CoT to answer the final personality inquiry. We prompt the LLM rating items based on the author's text, which simulates the process of human to complete personality tests through a multi-turn dialogue.

Bagby et al., 2016; Andrist et al., 2015; Hickman et al., 2022; Matz et al., 2017). For instance, personality aids clinical psychologists in gaining a better understanding of psychiatric disorders (Khan et al., 2005) and developing personalized treatment modalities (Bagby et al., 2016); it improves the human-robot interaction, particularly for socially assistive robots (Andrist et al., 2015).

Previous studies (Mehta et al., 2019; Yang et al., 2021b, 2022) on text-based personality detection have focused on training or fine-tuning specific models. However, their performance is significantly limited by the quality and quantity of training data. The emergence of large language models (LLMs), such as GPT-3 (Brown et al., 2020), InstructGPT (Ouyang et al., 2022), and LLaMA (Touvron et al., 2023), has recently demonstrated impressive in-context learning (ICL) ability, in which LLMs

---
*Corresponding authors.

make predictions solely based on designed prompts or instructions without any parameter modifications, leading to a new paradigm in the NLP community. Building upon these strengths, this study aims to investigate the ability of LLMs to perceive an author's personality from text, an aspect that has not been extensively explored previously.

Inspired by the process of human to complete self-assessment personality tests, we approach personality detection as a multi-step reasoning task since psychologists often use a series of assessments to measure an individual's personality. To perform complex reasoning, a recently technique named chain-of-thought (CoT) has been developed for solving math word problems (Cobbe et al., 2021; Patel et al., 2021) by generating intermediate reasoning steps (Wei et al., 2022b; Kojima et al., 2022; Zhang et al., 2022b). In our scenario, we argue that the items in psychological questionnaires can be considered as a set of rigorous reasoning steps. Consequently, we propose a novel personality reasoning framework, **PsyCoT**, which utilizes psychological questionnaires as CoT. Figure 1 illustrates the framework of PsyCoT, which mimics a human to complete personality test through a multi-turn dialogue. Specifically, we prompt the LLM (e.g., GPT-3.5[1]) to act as an AI assistant whose task is to rate[2] a given item based on the author's text. At each turn, we sample an item from the psychological questionnaire and present it to the assistant. The AI assistant then returns a specific score for the item. Once all the items have been rated, the AI assistant selects a definitive personality trait based on the historical rating results. Additionally, these scores can be aggregated using rules defined by questionnaires to yield an overall score, which severs as a double check and provides confidence for the chosen personality trait.

To evaluate the effectiveness of PsyCoT, we conduct extensive experiments on two benchmark datasets (i.e., Essays and Kaggle) that employ different personality taxonomies (i.e., Big Five (Goldberg, 1990) and MBTI (Myers-Briggs, 1991)). Experimental results show that PsyCoT significantly increases the zero-shot performance of GPT-3.5 in personality detection. For instance, PsyCoT outperforms the standard prompting by 4.23/10.63 points in average F1 on the two datasets. Moreover, in the Essays dataset, PsyCoT surpasses fine-tuned methods on most personality traits, demonstrating its competitiveness within the fine-tuning paradigm. Furthermore, ablation studies and analysis indicate that the form of multi-turn dialogue helps to achieve a more accurate reasoning than the single-turn dialogue, and PsyCoT exhibits strong robustness when faced perturbation in option order.

Our work is the first to explore the ability of LLMs in detecting personality. The proposed PsyCoT incorporates a well-designed questionnaire as CoT steps to facilitate the reasoning of LLM. We highlight that GPT-3.5 yields comparable performance to some fine-tuning methods by equipped with PsyCoT. Besides, PsyCoT offers a fresh perspective of using well-designed questionnaires to design prompts in the era of LLMs.

## 2 Related Work

**Personality Detection** In early stage, Pennebaker et al. (2001) developed Linguistic Inquiry and Word Count (LIWC) to extract psycholinguistic features from text, which has been used for feature engineering in machine learning models (Cui and Qi, 2017; Amirhosseini and Kazemian, 2020). However, these feature engineering-based methods have limitations in extracting implicit semantic features. Recently, several studies have proposed unbiased user representations for personality detection. For instance, Yang et al. (2021a, 2022) addressed the post-order problem in encoding posts by introducing a multi-document Transformer and a unordered dynamic deep graph network, respectively. Zhu et al. (2022) constructed a fully connected post graph for each user and developed CGTN to consider correlations among traits. Another body of research incorporates additional knowledge into the model. For example, Yang et al. (2021b) proposed TrigNet, which constructs a heterogeneous tripartite graph using LIWC and utilizes flow-GAT to operate on this graph. Yang et al. (2021b) introduced PQ-Net, which incorporates psychological questionnaires as additional guidance to capture item-relevant information from contextual representations. Despite their successes, these methods are rely on a data-driven approach to train the model to capture implicit personality cues, which are differ from our study as we specifically explore the zero-shot performance of the LLM in personality detection.

**LLMs and Prompting Methods** Recent years have witnessed an increasing interest in LLMs,

---

[1]https://platform.openai.com/docs/models/gpt-3-5

[2]We follow the original rating rules defined by psychological questionnaires.

such as GPT-3 (Brown et al., 2020), FLAN (Wei et al., 2022a), OPT (Zhang et al., 2022a), PaLM (Chowdhery et al., 2022), and LLaMA (Touvron et al., 2023), due to their impressive zero-shot generalization across various tasks. With techniques like instruction tuning (Wei et al., 2022a) and reinforcement learning with human feedback (RLHF) (Ouyang et al., 2022), ChatGPT[3] demonstrates remarkable alignment capabilities when following human instructions. Consequently, to leverage the potential of LLMs in downstream tasks, several works have focused on carefully designing prompts manually (Hendy et al., 2023) or automatically (Gao et al., 2021; Zhou et al., 2022b). Another approach that has gained attention is the Chain-of-Thought (CoT), which explicitly guide LLMs in generating reasoning steps. Wei et al. (2022b) initially introduced the few-shot-CoT, which utilized reasoning steps as demonstrations by crafting few-shot exemplars, resulting in significant improvements in complex math tasks. Building on this work, other studies have proposed various variants of CoT, including Zero-shot CoT (Kojima et al., 2022), Auto-CoT (Zhang et al., 2022b), Least-to-most prompting (Zhou et al., 2022a), and Synthetic prompting (Shao et al., 2023). Unlike most of these works that focus on how to select few-shot exemplars, we aim to use psychological questionnaire as a explicitly CoT to facilitate reasoning.

## 3 Psychological Questionnaire as CoT

For the current generative LLM, personality detection can be formulated as a reading comprehensive task. Formally, given a set of texts $X = \{x_1, x_2, \ldots, x_n\}$ written by an individual, the goal is to determine the correct personality option $Y = \{y^t\}_{t=1}^T$ about this individual based on $X$. We can achieve the goal by designing the appropriate prompt $P = \{D, I\}$, in which $D$ represents the task description prompt that informs the LLM about the task's definition, and $I$ represents the inference prompt that pushes the LLM to select the desired personality trait in a specified format.

### 3.1 Standard Prompting

We first describe the standard prompting, which infers the results directly from the given input text $X$ under prompted by $D$ and $I$:

$$\hat{y}^i = \text{LLM}\left(D, X, I\right) \qquad (1)$$

---

[3]https://chat.openai.com/

---

**Algorithm 1:** PsyCoT Prompting.

**Input:** LLM: LLM $(\cdot)$; Author's text: $X$;
      Task description prompt: $D$;
      Inference prompt: $I$; $K$ Reasoning
      prompts: $R = \{r_k\}_{k=1}^K$
**Output:** the personality trait: $\hat{y}^i$

1: Initialize a dialogue history $H = [null]$
2: **for** $k = 1$ to $K$ **do**
3:    Get $k$-th rating result $a_k$ under $r_k$:
     $a^k = \text{LLM}\left(D, X, H, r_k\right)$
4:    Append the $r_k$ and $a_k$ into $H$:
     $H = [r_1, a_1, \ldots, r_k, a_k]$
5: **end for**
6: Infer the personality trait:
     $y^i = \text{LLM}\left(D, X, H, I\right)$
7: **return** $\hat{y}^i$

---

where $\hat{y}^i$ is the inferred personality trait. We show an example of the standard prompting at the top of Figure 2. The initial paragraph introduces a task description prompt $D$, where we specify the role of LLM as an AI assistant specializing in text analysis. Its objective is to predict an individual's personality trait based on their written text. The subsequent paragraph contains the text written by the author, which will be analyzed by the LLM. Lastly, the final paragraph depicts the inference prompt $I$, which includes the provided options and instructions for the LLM to generate the choice using a fixed format. The standard prompting relies solely on the LLM to comprehend the concept of personality and determine how to evaluate it.

### 3.2 PsyCoT Prompting

While standard prompting techniques have demonstrated effectiveness across various tasks (Sanh et al., 2022; Wang et al., 2023), they fall short when confronted with the complex task of personality detection. This limitation arises from the fact that texts authored by individuals are often *topic-agnostic*, lacking explicit references to their personality traits. Motivated by self-assessment personality tests (Digman, 1990; Myers-Briggs, 1991), we introduce PsyCoT prompting, which utilizes items from questionnaires as a chain-of-thought (CoT) framework to enhance reasoning capabilities.

The example of PsyCoT utilizing the 44-item Big Five Inventory (John et al., 2008) is depicted at the bottom of Figure 2. In comparison to standard prompting, PsyCoT mimics the process of

```
┌─ Standard Prompting ──────────────────────────────────────────────────┐
│                                                                        │
│ <Task description prompt D> You are an AI assistant who specializes in │
│ text analysis. You will complete a text analysis task. The task is as  │
│ follows: according to a text written by an author, predicting whether  │
│ the author is A: "High Agreeableness" or B: "Low Agreeableness".       │
│ <Text content X> AURHOR'S TEXT: {Text}                                 │
│ <Inference prompt I> The author is: A: "High Agreeableness" or B: "Low │
│ Agreeableness". Provide a choice in the format: "CHOICE: <A/B>" and do  │
│ not give the explanation.                                              │
│                                                                        │
└────────────────────────────────────────────────────────────────────────┘

┌─ PsyCoT Prompting ────────────────────────────────────────────────────┐
│                                                                        │
│ <Task description prompt D> You are an AI assistant who specializes in │
│ text analysis and I am Human. We will complete a text analysis task    │
│ together through a multi-turn dialogue. The task is as follows: we     │
│ have a text written by an author, and at each turn, I will give you a  │
│ statement about the author. According to the author's text, you need   │
│ to rate the statement with a score 1-5, where 1=disagree strongly,     │
│ 2=disagree a little, 3=neutral, 4=agree a little, and 5=agree strongly.│
│ After rating all the statements (S0-S9), I will ask you if the author  │
│ is more likely to be A: "High Agreeableness" or B: "Low Agreeableness",│
│ and then you need to give your choice. Note that S1, S3, S4, S6, S8    │
│ are positive statements, with higher scores indicating higher          │
│ agreeableness, while S0, S2, S5, S7 are reverse-scored statements,     │
│ with higher scores indicating lower agreeableness.                     │
│ <Text content X> AUTHOR'S TEXT: {Text}                                 │
│ ┌ ─ ─ ─ ─ ─ ─ ─ ─ ─ ─ ─ ─ ─ ─ ─ ─ ─ ─ ─ ─ ─ ─ ─ ─ ─ ─ ─ ─ ─ ─ ─ ─ ┐│
│ Human: S0: The author is original, comes up with new ideas. Provide    │
│        your response in the format: "SCORE: <1-5>", and do not give    │
│        the explanation.                                                │
│ └ ─ ─ ─ ─ ─ ─ ─ ─ ─ ─ ─ ─ ─ ─ ─ ─ ─ ─ ─ ─ ─ ─ ─ ─ ─ ─ ─ ─ ─ ─ ─ ─ ┘│
│ Assistant: SCORE: 2                                                    │
│ …                                                                      │
│ (after rating all the statements)                                      │
│ Human: <Inference prompt I> According to above scores, the author is:  │
│ A: "High Agreeableness" or B: "Low Agreeableness". Provide a choice in │
│ the format: "CHOICE: <A/B>" and do not give the explanation.           │
│ Assistant:                                                             │
│                                                                        │
└────────────────────────────────────────────────────────────────────────┘
```

Figure 2: Comparison of Standard Prompting (Top) and our PsyCoT Prompting (Bottom). In PsyCoT, the dotted box indicates a reasoning step derived from a psychological questionnaire. Unlike Standard Prompting, which directly prompts LLM to output the personality preference, PsyCoT incorporates multiple reasoning steps through interactions with the LLM, guiding the LLM to infer personality in a more reasonable manner.

self-assessment personality tests and enhances standard prompting in the following key aspects: (1) We modify the task description $D$ by instructing the LLM to rate each statement (item in the questionnaire). (2) We include in $D$ a description of the rating rules for the questionnaire, encompassing the scoring system (e.g., "*1=disagree strongly, 2=disagree a little, 3=neutral, 4=agree a little, and 5=agree strongly*") and highlighting the significance of reversed statements. (3) Most importantly, we introduce $K$ reasoning steps $R = \{r_k\}_{k=1}^{K}$ prior to accessing the personality trait $\hat{y}^i$ via multi-dialogue, guiding the LLM in inferring the personality with a more reasonable manner. The overall progress of PsyCoT is described in Algorithm 1, including step by step reasoning in Line 2-4. We provide complete dialogue records in Appendix B.

A by-produce of PsyCoT Prompting is the rating results $[a_1, a_2, \ldots, a_K]$, which can be aggregated into an overall score like the self-assessment personality tests. For example, the overall score $s^i$ in 44-item Big Five Inventory is computed as:

$$s^i = \frac{1}{K} \sum_{k=1}^{K} s_k \qquad (2)$$

where $s_k$ is the transition score, which defined as:

$$s_k = \begin{cases} 6 - a_k & r_k \in R_s \\ a_k & \text{otherwise} \end{cases} \qquad (3)$$

where $R_s$ is the set of reversed statements. Since positively correlated with $y_i$, the overall score $s^i$ can serve as a double check and provide confidence for the inferred $\hat{y}^i$.

**Discussion** PsyCoT employs a multi-turn approach to rating questionnaire items, as opposed to providing simultaneous ratings for all items within a single turn. There are two main reasons. Firstly, by rating each item individually during each turn, we can maintain a higher level of focus for the LLM, resulting in more accurate outcomes. Secondly, there are inherent correlations between the items. For instance, statements such as "*The author*

*is reserved.*" and "*The author tends to be quite.*" exhibit a high level of correlation. Consequently, incorporating the historical rating results when evaluating the current item within a multi-turn structure will enhance the consistency of the obtained results. We provide more investigations in Section 4.5.

# 4 Experiment

## 4.1 Datasets

We evaluate PsyCoT on two public datasets: Essays (Pennebaker and King, 1999) and Kaggle[4]. The Essays comprises 2468 anonymous texts written by volunteers in a strict environment. These volunteers were instructed to write whatever came to their mind within a short time-frame. Each essay is associated with the author's Big Five personality traits, namely Agreeableness, Conscientiousness, Extraversion, Neuroticism, and Openness. The authors assessed their own traits using a 44-item Big Five Inventory (John et al., 1991), which is the same questionnaire we employ in PsyCoT. To ensure fair comparisons with fine-tuned methods, we randomly divided the dataset into training, validation, and testing with proportions of 80-10-10, respectively. For prompt-based methods, we evaluate their zero-shot performance on the test set.

The Kaggle dataset is collected from Personality-Cafe[5], a forum where people share their personality traits and engage in discussions about personality-related topics. The Kaggle comprises a total of 8675 user's data, each consisting of 45-50 posts. The used personality taxonomy is MBTI (Myers-Briggs, 1991), which divides personality into four dimensions: Introversion or Extroversion (I/E), Sensing or iNtuition (S/N), Thinking or Feeling (T/F), and Perception or Judging (P/J). Limited by API resources, we randomly selected 200 samples form the test set to evaluate PsyCoT and baselines.

For the Essay dataset, the average token count per sample is 762.38, whereas for Kaggle, it stands at 1632.38. More details of the two datasets and the used questionnaires are provided in Appendix C and Appendix D, respectively.

## 4.2 Baselines

In our experiments, we adopt the following previous methods as baselines.

**LIWC+SVM** (Tighe et al., 2016): This shallow method firstly extracts psycholinguistic features

using LIWC (Pennebaker et al., 2001), and then applies SVM as the classifier.

**TF-IDF+SVM** (Cui and Qi, 2017): This method is similar to LIWC+SVM, but it extracts features using TF-IDF.

**W2V+CNN** (Rahman et al., 2019) and **Glove+LSTM** (Sun et al., 2018): These two methods are non-pretrained models. The former encodes the context using the word2vec algorithm and then applies CNN (Chen, 2015) to obtain the context representation. The latter uses Glove (Pennington et al., 2014) for word embeddings and then utilizes LSTM (Hochreiter and Schmidhuber, 1997) to encode the context.

**Regression** (Park et al., 2015): This method trains a regression model with two regression scores 0 and 1. The same quantile discretization method from Ganesan et al. (2023) is then used to convert the test set scores into categorical labels.

**BERT** (Devlin et al., 2019) and **RoBERTa** (Liu et al., 2019): These two models are fine-tuned and utilize "bert-base-cased" and "roberta-base" as backbones, respectively. For Essays, they encode the context directly, while for Kaggle, they encode each post and then aggregate post representations via mean pooling.

**SN+Attn** (Lynn et al., 2020): This method adopts a hierarchical attention network to generate the user representation. Following Yang et al. (2021b), the pre-trained BERT is utilized as a post-encoder to ensure fair comparisons.

**TrigNet** (Yang et al., 2021b): TrigNet constructs a tripartite graph with psycholinguistic knowledge in LIWC to fuse posts.

**DDGCN** (Yang et al., 2022): DDGCN is the latest SOTA method in the Kaggle dataset, which firstly encodes each post using a domain-adapted BERT, and then aggregates the posts in a disorderly manner by a dynamic deep graph network.

For prompt-based methods, in addition to standard prompting, we adopt **Zero-shot-CoT** (Kojima et al., 2022), which inserts a reasoning step with "*Let's think step by step.*" before accessing the final personality via the inference prompt.

## 4.3 Implementation Details

In this study, we simplify multi-label personality detection into multiple binary classification tasks.[6] For the prompt-based methods, we request the GPT-3.5 API (gpt-3.5-turbo-0301) to obtain results,

---

[4]https://www.kaggle.com/datasnaek/mbti-type
[5]http://personalitycafe.com/forum

[6]More details are provided in Appendix A

| Methods | AGR | | CON | | EXT | | NEU | | OPN | | Average | |
|---|---|---|---|---|---|---|---|---|---|---|---|---|
| | Acc. | F1 | Acc. | F1 | Acc. | F1 | Acc. | F1 | Acc. | F1 | Acc. | F1 |
| LIWC+SVM[†] | 51.78 | 47.50 | 51.99 | 52.00 | 51.22 | 49.20 | 51.09 | 50.90 | 54.05 | 52.40 | 52.03 | 50.40 |
| Regression | 50.96 | 51.01 | 54.65 | 54.66 | 55.06 | 55.06 | 57.08 | 57.09 | 59.51 | 59.51 | 55.45 | 55.47 |
| W2V+CNN[†] | - | 46.16 | - | 52.11 | - | 39.40 | - | 58.14 | - | 59.80 | - | 51.12 |
| BERT | 56.84 | 54.72 | 57.57 | 56.41 | 58.54 | 58.42 | 56.60 | 56.36 | 60.00 | 59.76 | 57.91 | 57.13 |
| RoBERTa | 59.03 | 57.62 | 57.81 | 56.72 | 57.98 | 57.20 | 56.93 | 56.80 | 60.16 | 59.88 | 58.38 | 57.64 |
| Standard | 59.11 | 57.98 | 57.49 | 49.55 | 59.92 | 54.39 | 61.13 | 59.95 | 55.87 | 49.11 | 58.70 | 54.20 |
| Zero-shot-CoT | 58.94 | 58.09 | 55.14 | 42.49 | 57.55 | 55.63 | 57.49 | 54.63 | 58.78 | 54.40 | 57.58 | 53.05 |
| PsyCoT | 61.13 | 61.13 | 59.92 | 57.41 | 59.76 | 59.74 | 56.68 | 56.58 | 60.73 | 57.30 | 59.64 | 58.43 |

Table 1: Overall results of PsyCoT and baselines on the Essays dataset. We use Accuracy(%) and Macro-F1(%) as metrics. The symbol † means results directly taken from the original papers. Best results are listed in bold and the second best results are shown with underline.

| Methods | I/E | | S/N | | T/F | | J/P | | Average | |
|---|---|---|---|---|---|---|---|---|---|---|
| | Acc. | F1 | Acc. | F1 | Acc. | F1 | Acc. | F1 | Acc. | F1 |
| TF-IDF+SVM | 71.00 | 44.94 | 79.50 | 46.38 | 75.00 | 74.25 | 61.50 | 58.59 | 71.75 | 56.04 |
| Regression | 61.34 | 64.00 | 47.10 | 54.50 | 76.34 | 76.50 | 65.58 | 66.00 | 62.59 | 65.25 |
| Glove+LSTM | 72.50 | 62.04 | 80.50 | 52.78 | 74.00 | 73.23 | 62.50 | 60.31 | 72.38 | 62.09 |
| BERT | 77.30 | 62.50 | 84.90 | 54.04 | 78.30 | 77.93 | 69.50 | 68.80 | 77.50 | 65.82 |
| SN+Attn | - | 65.43 | - | 62.15 | - | 78.05 | - | 63.92 | - | 67.39 |
| RoBERTa | 77.10 | 61.89 | 86.50 | 57.59 | 79.60 | 78.69 | 70.60 | 70.07 | 78.45 | 67.06 |
| TrigNet | 77.80 | 66.64 | 85.00 | 56.45 | 78.70 | 78.32 | 73.30 | 71.74 | 78.70 | 68.29 |
| DDGCN | 78.10 | 70.26 | 84.40 | 60.66 | 79.30 | 78.91 | 73.30 | 71.73 | 78.78 | 70.39 |
| Standard | 52.00 | 51.52 | 47.00 | 43.76 | 68.00 | 67.68 | 55.50 | 55.41 | 55.63 | 54.59 |
| Zero-shot-CoT | 76.50 | 64.27 | 83.50 | 55.16 | 72.50 | 71.99 | 57.50 | 53.14 | 72.50 | 61.14 |
| PsyCoT | 79.00 | 66.56 | 85.00 | 61.70 | 75.00 | 74.80 | 57.00 | 57.83 | 74.00 | 65.22 |

Table 2: Overall results of PsyCoT and baselines on the Kaggle dataset.

which is currently the most popular and forms the foundation of ChatGPT. We set the temperature as 0 to ensure deterministic outputs. In the case of Essays, we limited the maximum tokens for the author's text to 3200, while for Kaggle, we restricted it to 80 tokens per post. For the fine-tuning based methods, we set the learning rate to 2e-5, and report their test performance (using same data as prompt-based methods) by averaging the results of five runs. The evaluation metrics employed in our study include Accuracy and Macro-F1 score.

## 4.4 Overall Results

The overall results of PsyCoT and several baselines on Essays are listed in Table 1. Data-driven baselines comprise three types of methods: shallow model (LIWC+SVM and Regression), non-pretrained model (W2V+CNN), and fine-tuned models (BERT and RoBERTa). The prompt-based baselines include Standard prompting and Zero-shot-CoT.

There several key observations from these results. *First*, PsyCoT outperforms the baselines on most personality traits, even surpassing the fine-tuned models. Specifically, PsyCoT enhances standard prompting with an average increase of 0.94/4.23 points in accuracy and macro-F1. *Second*, PsyCoT performs worse than standard prompting on the Neuroticism trait. Further analysis reveal that this discrepancy may be attributed to dataset bias. The Essays contains a significant amount negative emotions expressed by authors, which misleads the LLM into assigning high scores for statements such as "*The author worries a lot.*", "*The author is depressed, blue.*", and "*The author can be moody.*". *Third*, although includes a reasoning step with "*Let's think step by step.*", Zero-shot-CoT does not consistently improve the performance of standard prompting. Our investigation shows that Zero-shot-CoT often fails to guide the reasoning process correctly, resulting in meaningless responses like "*Sure, what would you like to start with?*". In contrast, PsyCoT directly constructs reasoning steps with the help of questionnaires.

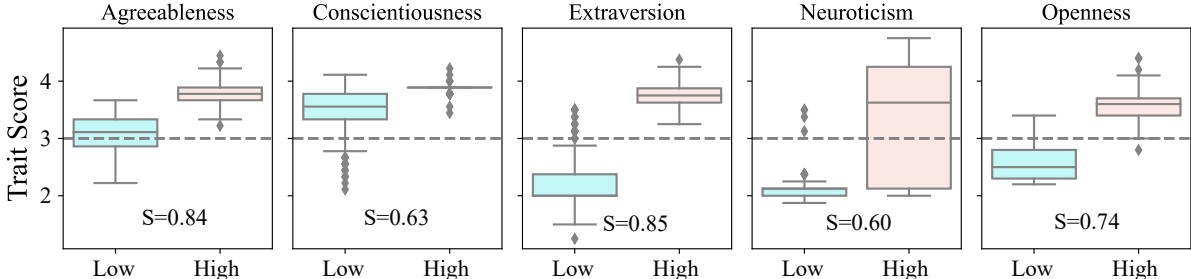

Figure 3: Distributions of the trait scores across five personalities in the Essays dataset. The dashed line represents the neutral value defined by the questionnaire (i.e., 3=Neutral). The value of "S" displayed in figures indicate the Spearman's Coefficient between trait scores and the personality dimensions. A value closer to 1 suggests a stronger positive correlation. Results demonstrate a strong positive correlation between trait scores and personality type, particularly in Agreeableness, Extraversion, and Openness.

Table 2 presents additional results on the Kaggle dataset. The obeservations are threefold. *First*, PsyCoT achieves the highest performance among prompt-based methods with 18.37/10.63 points improvement in accuracy and macro-F1 over standard prompting, and 1.50/4.08 points improvement over Zero-shot-CoT. *Second*, compared to fine-tuned models, PsyCoT achieves a performance very close to that of BERT, while still falls behind the current SOTA method DDGCN by a non-negligible margin. To explore the underlying reasons, we analyze the textual input and find that users in the Kaggle dataset directly discuss the MBTI and frequently employ abbreviations such as "Fi," "Fe," "Ni," "Te," and "IxxP" to describe personality traits. These abbreviations hold potential as strong features for data-driven methods, but for zero-shot methods, they may be insufficient for capturing the association between these abbreviations and personalities. *Third*, despite being weaker than the fine-tuned methods on the Kaggle, prompt-based methods have the advantages of not requiring the collection of training data and retraining of models to accommodate changes in personality types and application scenarios (e.g., Essays and Kaggle should be quite different in terms of the text style: freely-written text vs. discussion posts.)

### 4.5 Ablation Study

Table 3 presents several ablation results of PsyCoT to facilitate more comprehensive investigations.

**Single-turn**   PsyCoT adopts a multi-turn dialogue structure to facilitate reasoning. This structure allows the LLM to concentrate on a particular item, enhancing the accuracy of its reasoning process. To verify its effectiveness, we test another alternative

| Methods | Accuracy | Δ | Macro-F1 | Δ |
|---|---|---|---|---|
| PsyCoT | 59.64 | - | 58.43 | - |
| *r/w* single-turn | 59.27 | 0.37↓ | 56.00 | 2.43↓ |
| *r/w* TIPI | 58.46 | 1.18↓ | 54.73 | 3.70↓ |
| *r/w* Mini-IPIP | 59.19 | 0.45↓ | 57.42 | 1.01↓ |

Table 3: Abalation results of PsyCoT in average Accuracy and Macro-F1 on the Essays dataset, where "Δ" indicates the corresponding performance change, and *r/w* means "replace with".

approach, which involves presenting all the items within a single-turn dialogue and instructing the LLM to assign ratings to all of them simultaneously. From Table 3, we can observe that reasoning with a single-turn dialogue deteriorates the performance, demonstrating that PsyCoT with multi-turn for reasoning is preferable.

**Other Questionnaire**   We utilizes the 44-item Big Five Inventory (John et al., 1991) for operating PsyCoT in Essays. However, alternative questionnaires also can be applied in PsyCoT. Therefore, we have included other widely used brief measures, such as TIPI (10-item) (Gosling et al., 2003) and Mini-IPIP (20-item) (Donnellan et al., 2006), to examine the impact of different questionnaires. Appendix D presents a comparison between these three inventories. It is evident that the 44-item inventory offers a broader range of aspects for evaluating an individual's personality, making it a potentially more effective CoT for guiding reasoning. The results in Table 3 further confirm that the TIPI is insufficient in eliciting the desired level of reasoning, as it leads to a significant decline in performance. While the 20-item scale is better than 10-item scale, but still inferior to the 44-item scale. This experiment underscores the crucial of

selecting appropriate questionnaires for PsyCoT.

# 5 Analysis

## 5.1 Correlation Analysis

In this study, we explicitly introduce the psychological questionnaire as CoT, which guides the LLM in assessing the author's personality by rating each item. Consequently, as defined by Eq.(2), these ratings can be computed into an overall score (we refer to trait score) that reflects the strength of a specific personality trait. To investigate the correlation between the trait score and the chosen personality trait, we firstly visualize the distributions of trait scores across five different dimensions in the Essays dataset, as shown in Figure 3.

The observations yield two main findings. *First*, in general, there is a significant difference in the distribution between low and high traits, and the trait scores exhibit a strong positive correlation with the personality types, particularly in Agreeableness, Extraversion, and Openness. *Second*, the distribution of scores for Extraversion and Openness can be distinguished by the dashed line, illustrating that the LLM's ratings for these traits align with the criteria of the questionnaire. On the other three traits, however, the distributions intersect the dashed line and shift to one side, suggesting potential bias in the LLM's ratings of the questionnaires. We further apply Spearman's Coefficient (Sedgwick, 2014) to quantitatively measure the correlations. From Figure 3, the trait scores for Agreeableness, Extraversion, and Openness have a higher correlation ($> 0.70$) with the chosen personality type, which validates our observations.

## 5.2 Statistical Tests

To test the statistical significance of our method, we conduct a comparison between the Standard prompt and PsyCot using the Essays dataset, and the F1 results are presented in the Table 4. Each point represents the average of 5 runs. The T-test analysis indicates that our improvements are statistically significant, with p-values of less than 0.05 for AGR, less than 0.001 for CON, EXT, and OPN.

| Methods | AGR | CON | EXT | NEU | OPN |
|---------|-----|-----|-----|-----|-----|
| Standard | 58.17 | 49.52 | 53.96 | 59.46 | 59.46 |
| PsyCoT | 60.34⋄ | 57.44⋆ | 59.23⋆ | 59.46 | 57.88⋆ |

Table 4: Significant tests on the Essays dataset. ⋄ and ⋆ mean $p < 0.5$ and $p < 0.001$, respectively.

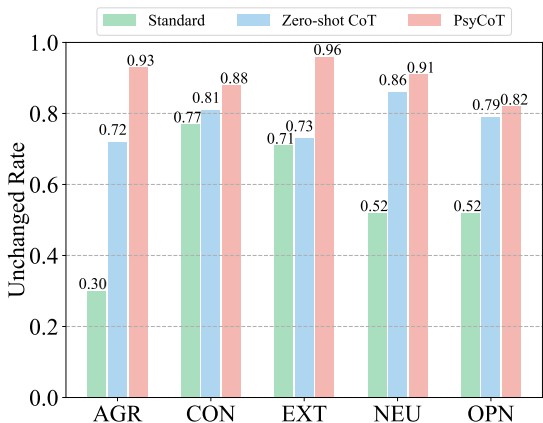

Figure 4: The robustness testing results for three prompt-based methods on the five traits. PsyCoT demonstrates the highest unchanged rate, indicating its robustness in handling option order perturbations.

## 5.3 Robustness Testing

A notorious drawback of LLMs is its vulnerability to minor variations in prompts, such as changes in the order of options within choice questions. Intuitively, PsyCoT incorporates questionnaire to facilitate reasoning, potentially improving the robustness of LLMs. To verify this hypothesis, we conduce an experiment where we swap the option orders in both the task description prompt $D$ and inference prompt $I$ (e.g., A: "High Agreeableness" or B: "Low Agreeableness" is exchanged to A: "Low Agreeableness" or B: "High Agreeableness"), and re-test the prompt-based methods. We measure the unchanged rate of $\hat{y}^i$ across 100 samples. The results from the Essays dataset are presented in Figure 4. PsyCoT achieves the highest unchanged rate and significantly superiors other methods, demonstrating that the inclusion of the questionnaire enhances its robustness.

## 5.4 Impact of Post Order

The Kaggle dataset contains a set of posts for each user. These posts are concatenated in sequence to create a long document, which serves as the input $X$. However, Yang et al. (2021a, 2022) have demonstrated that encoding posts sequentially is order-sensitive for the fine-tuned models. To investigate whether the prompt-based methods are also influenced by post order, we randomly disrupt the post orders and re-test the prompt-based methods using 100 samples. Similar to Section 5.3, the unchanged rate is used for evaluation. As shown in Figure 5, shuffling post orders leads to noticeable changes in the predictions across several personal-

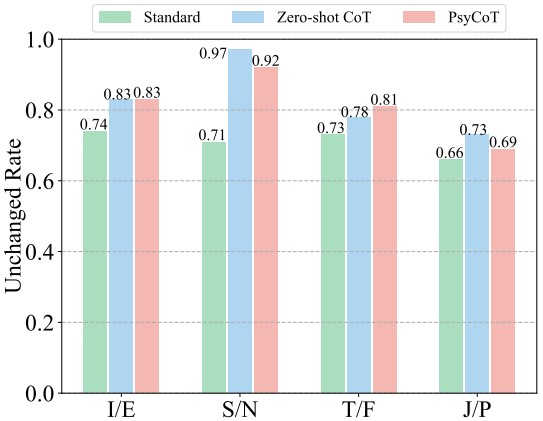

Figure 5: Results of the study on post orders. The three prompt-based methods are influenced by post orders for most of personality traits.

ity traits, including PsyCoT. This observation highlights that the order of posts remains an important issue for the LLM.

## 6  Conclusion

In this paper, we propose a novel method, PsyCoT, for zero-shot personality detection. PsyCoT prompts the LLM to engage in reasonable personality reasoning. Unlike the standard prompting approach that directly requests the LLM to infer the answer, PsyCoT draws inspiration from self-assessment personality tests that evaluate an individual's personality through well-designed questionnaires. PsyCoT introduces items from the questionnaire as a set of rigorous CoT. It guides the LLM to evaluate each item based on the author's text and then infer the final personality trait. Extensive experiments on two benchmark datasets demonstrate that PsyCoT outperforms Standard prompting and Zero-shot-CoT approaches by a large margin. It also achieves comparable performance to some fine-tuned models. This work represents the first effort to leverage psychological questionnaires to elicit the LLM's reasoning abilities, providing a new perspective for exploring the use of questionnaires in the era of LLMs.

## Acknowledgements

This work was supported by the National Natural Science Foundation of China (No. 62176270), the Guangdong Basic and Applied Basic Research Foundation (No. 2023A1515012832), and 2023 Tencent Rhino-Bird Research Elite Program.

## Limitations

The potential limitations of PsyCoT are discussed below. Firstly, PsyCoT utilizes a multi-turn dialogue approach to rate items from the questionnaire. Although this approach surpasses the single-turn method (as reported in Section 4.5), it does result in more API requests. Secondly, the performance of PsyCoT is heavily influenced by the selected questionnaire, necessitating additional effort in finding the most suitable one. Thirdly, PsyCoT has only been applied to one language, and there is a need to explore its application in a broader range of languages.

## Ethics Statement

Text-based personality detection is a long history research area within psycholinguistics. We clarify that the purpose of this study is to explore and improve the ability of the LLM in this particular task, rather than creating a tool that invades privacy. The Essays and Kaggle datasets used in our study are public available and all user information has been anonymized. We have strictly adhered to the data usage policy throughout our research. However, it is crucial to acknowledge that the misuse of this technology may pose ethical risks. We state that any research or application stemming from this study is solely permitted for research purposes, and any attempt to exploit this technology to covertly infer individuals' personality traits or other sensitive characteristics is strictly prohibited.

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

## A   Appendix: Form of the Task

Although the original tasks of the Big 5 are in the regression form, the Essays dataset is further collected and processed into the binary label format (Y/N), without releasing the original scores from the raters. Consequently, we have framed this task as a classification problem, designating 'High' as 'Y' and 'Low' as 'N.' Besides, our initial study indicates that simplifying multi-label personality detection into multiple binary classification tasks leads to performance improvements (the average results on standard prompt are reported in Table 5). Hence, this approach was chosen for our experiments.

| Forms | Accuracy | Macro-F1 |
|---|---|---|
| Multi-label | 51.74 | 46.85 |
| Five binary (ours) | **58.70** | **54.20** |

Table 5: Comparison results of different forms.

## B   Appendix: Dialogue Record

To better understand our method, we provide two complete records in Figure 6 and Figure 7, respectively. These records demonstrate how PsyCoT predicts Big Five and MBTI personality traits.

## C   Appendix: Details of Datasets

We provide the statistics of Essays and Kaggle in Table 6.

| Dataset | Types | Train | Validation | Test |
|---|---|---|---|---|
| Essays | AGR | 1051 / 922 | 132 / 115 | 127 / 120 |
| | CON | 1019 / 954 | 121 / 126 | 113 / 134 |
| | EXT | 1019 / 954 | 122 / 125 | 135 / 112 |
| | NEU | 985 / 988 | 121 / 126 | 127 / 120 |
| | OPN | 1011 / 962 | 127 / 120 | 133 / 114 |
| Kaggle | I / E | 4011 / 1194 | 1326 / 409 | 152 / 48 |
| | S / N | 610 / 4478 | 222 / 1513 | 27 / 173 |
| | T / F | 2410 / 2795 | 791 / 944 | 85 / 115 |
| | P / J | 3096 / 2109 | 1063 / 672 | 122 / 78 |

Table 6: Statistics of the Essays and Kaggle datasets.

## D   Appendix: Details of Questionnaires

In this study, the 44-item Big Five Inventory and Myers-Briggs Type Indicator are used for our methods. The details of items are listed in Table 7 and Table 10. Note that the original items in Myers-Briggs Type Indicator do not include the option "Not sure whether A or B". We include this option to ensure that the LLM has the ability to indicate uncertainty in cases where the provided information is insufficient for making a definitive choice. Besides, in Section 4.5, we utilize TIPI and Mini-PIPI for comparison. The details of these two scales are listed in Table 8 and Table 9, respectively.

```
# Record of Big Five

You are an AI assistant who specializes in text analysis and I am Human. We will complete a text
analysis task together through a multi-turn dialogue. The task as follows: we have a text written by
an author, and at each turn I will give you a statement about the author. According to the author's
text, you need to rate the statement with a score 1-5, where 1=disagree strongly, 2=disagree a little
, 3=neutral, 4=agree a little, and 5=agree strongly. After rating all the statements (S0-S8), I will
ask you if the author is A: "High Agreeableness" or B: "Low Agreeableness", and then you need to give
 your choice. Note that S1, S3, S4, S6, S8 are positive statements, with higher scores indicate
higher agreeableness, while S0, S2, S5, S7 are reverse-scored statements, with higher scores indicate
 lower agreeableness.
AUTHOR'S TEXT:
I'm pretty happy with my first week and a half of classes. I've met a lot of people. It means a lot
to me. I am a very quiet, sort of shy person. I was afraid of coming to UT because I'm not the best
at making friends. So far though, it hasn't been a problem. My big classes have been a little
intimidating because of there size. I used to have classes about 30 or less. Right now I am happy, UT
 won its first football game. I'm glad I was able to go. It was lots of fun. Now I feel a little
exhausted. I haven't really done all that much today. I actually got to sleep for 12 hours last night
. I could be exhausted because I am hungry. I think I'm pretty lucky. My parents sent me some food.
My brother brought it to Austin. He and I will share it. I'm already starting to miss home-cooked
meals. I think my 20 minutes are just about up. I probably fix a bite to eat, watch a little T. V.
and go to sleep.
Human: S0: "The author sometimes notices other people's faults." Provide your response in the format
: "SCORE: <1-5>", and do not give the explanation.
Assistant: SCORE: 3 (neutral)
Human: S1: "The author selflessly helps others." Provide your response only with the score.
Assistant: SCORE: 3 (neutral)
Human: S2: "The author sometimes argues with others." Provide your response only with the score.
Assistant: SCORE: 1 (disagree strongly)
Human: S3: "The author forgives others easily." Provide your response only with the score.
Assistant: SCORE: 4 (agree a little)
Human: S4: "The author trusts people easily." Provide your response only with the score.
Assistant: SCORE: 4 (agree a little)
Human: S5: "The author can be distant with others sometimes." Provide your response only with the
score.
Assistant: SCORE: 4 (agree a little)
Human: S6: "The author is kind to everyone." Provide your response only with the score.
Assistant: SCORE: 2 (disagree a little)
Human: S7: "The author can be rude sometimes." Provide your response only with the score.
Assistant: SCORE: 2 (disagree a little)
Human: S8: "The author likes to collaborate with others." Provide your response only with the score.
Assistant: SCORE: 5 (agree strongly)
Human: According to above scores, the author is more likely to be: A: "High Agreeableness" or B: "Low
 Agreeableness". Provide a choice in the format: "CHOICE: <A/B>" and do not give the explanation.
Assistant: CHOICE: A
```

Figure 6: A complete record that PsyCoT is applied to predict Big Five personality traits.

```
# Record of MBTI

You are an AI assistant who specializes in text analysis and I am Human. We will complete a text
analysis task together through a multi-turn dialogue. The task as follows: we have a set of posts
written by an author, nd at each turn I will give you a Question about the author. According to the
author\'s text, you need to choose the possible options. After opting all the choices, I will ask you
 if the author is A: "Extraversion" or B: "Introversion", and then you need to give your choice.
AUTHOR'S POSTS:
Post1: I dreamed a lot of weird stuff but generally my dream always sent me back to school for some
reason to meet someone I haven't met in a long while. Previously I dreamed of flying in a machine...;
 Post2: Maybe some headless dolls?; Post3: People like thank yous as much as the thumbs up icon so
keep sending them.; Post4: Try this. 717794; Post5: No more spam? Nuuuuuuuuuu; Post6: For some people
, life is a buffet. For others, their world is a set meal. LOL.; Post7: I currently desire pancakes.;
 Post8: Fi dom signature.; Post9: Neither heartlessness and indecision for not following your guts
will get you anywhere.; Post10: Hahahahahaha!!!; Post11: What makes you so certain that only s and
not s can be writers? Both are good with written words. Just a genuine question. :happy:; Post12:
Seems likely.; Post13: - If running the treadmill is not for you, you could either try the rowing
machine or if there is a public pool available, try swimming. They are relatively low-impact
exercises but would still give...; Post14: I was so sad I went for a long walk all the way to the gym
. Good thing though I didn't have to sign up for membership or anything. It was my first time there.
I was surrounded by old dude and...; Post15: Being an artist is not just about expressing yourself.
It's about trying to understand and connecting life in the way that would make you learn more about
yourself. Before one is an artist, one is a...; Post16: I only think of them as sentimental. Not
really conventional.; Post17: I'm hungry just looking at them..; Post18: You are a definite NF. Other
 than that, I May need to get to know you a little more..; Post19: It's almost 3 a.m. here and I've
read it as NT's domino club.; Post20: Most of my clothes function in such a way that at any moment, I
 can go for a jog, jump into a swimming pool and then proceed to bunjee jump off the building. The
fabric has to be stretchy and...; (Due to page limitations, we omit the rest of posts.)
Human: Q: "The author is usually: A: "A good mixer with gropus of people", B: "Quiet and reserved",
or C: "Not sure whether A or B"." Provide a choice ID in the format: "CHOICE: <A/B/C>", and do not
give the explanation.
Assistant: CHOICE: C
Human: Q: "Among the author's friends, the author is: A: "Full of news about everybody", B: "One of
the last to hear what is going on", or C: "Not sure whether A or B"."
Assistant: CHOICE: B
Human: Q: "The author tends to have: A: "A broad range of friendships with many different people", B:
 "Deep friendship with very few people", or C: "Not sure whether A or B"."
Assistant: CHOICE: C
Human: Q: "When the author is with a group of people, the author is usually: A: "Join in the talk of
the group", B: "Stand back and listen first", or C: "Not sure whether A or B"."
Assistant: CHOICE: B
Human: Q: "The author is: A: "Talk easily to almost anyone", B: "Find a lot to say only to certain
people or under certain conditions", or C: "Not sure whether A or B"."
Assistant: CHOICE: B
Human: Q: "In a large group, the author is more often: A: "Introduce others", B: "Get introduced", or
 C: "Not sure whether A or B"."
Assistant: CHOICE: B
Human: Q: "When the author meets the new people, the author tells what they are interested in: A: "
Right away", B: "Only after people to get to know the author", or C: "Not sure whether A or B"."
Assistant: CHOICE: B
Human: Q: "The author is usually: A: "Show their feelings freely", B: "Keep their feelings to
themselves", or C: "Not sure whether A or B"."
Assistant: CHOICE: B
Human: Q: "Which word is more suitable for the author: A: "QUIET", B: "HEARTY", or C: "Not sure
whether A or B"."
Assistant: CHOICE: A
Human: Q: "Which word is more suitable for the author: A: "RESERVED", B: "TALAKATIVE", or C: "Not
sure whether A or B"."
Assistant: CHOICE: A
Human: Q: "Which word is more suitable for the author: A: "CALM", B: "LIVELY", or C: "Not sure
whether A or B"."
Assistant: CHOICE: A
Human: According to above choices, the author is more likely to be: A: "Extraversion" or B: "
Introversion". Provide a choice in the format: "CHOICE: <A/B>" and do not give the explanation.
Assistant: CHOICE: B
```

Figure 7: A complete record that PsyCoT is applied to predict MBTI personality traits.

| Agreeableness | Conscientiousness |
|---|---|
| #1: The author sometimes notices other people's faults. | #1: The author does things carefully and completely. |
| #2: The author selflessly helps others. | #2: The author can be kind of careless. |
| #3: The author sometimes argues with others. | #3: The author is a good, hard worker. |
| #4: The author forgives others easily. | #4: The author isn't very organized. |
| #5: The author trusts people easily. | #5: The author tends to be lazy. |
| #6: The author can be distant with others sometimes. | #6: The author keeps working until things are done. |
| #7: The author is kind to everyone. | #7: The author does things quickly and carefully. |
| #8: The author can be rude sometimes. | #8: The author makes plans and sticks to them. |
| #9: The author likes to collaborate with others. | |

| Extraversion | Neuroticism |
|---|---|
| #1: The author is talkative. | #1: The author is depressed, blue. |
| #2: The author is reserved. | #2: The author is relaxed, handles stress well. |
| #3: The author is full of energy. | #3: The author can be tense. |
| #4: The author generates a lot of enthusiasm. | #4: The author worries a lot. |
| #5: The author tends to be quiet. | #5: The author is steady, not easily upset. |
| #6: The author has an assertive personality. | #6: The author can be moody. |
| #7: The author is sometimes shy, inhibited. | #7: The author stays calm in difficult situations. |
| #8: The author is outgoing, sociable. | #8: The author gets nervous easily. |

| Openness | |
|---|---|
| #1: The author is original, comes up with new ideas. | |
| #2: The author is inquisitive about many things. | |
| #3: The author is ingenious, a deep thinker. | |
| #4: The author has a strong imagination. | |
| #5: The author is good at innovation. | |
| #6: The author values artistic. | |
| #7: The author works routine. | |
| #8: The author often reflects on himself. | |
| #9: The author has ordinary artistic interests. | |
| #10: The author is skilled in art, music, or literature. | |

Table 7: Items in 44-item Big Five Inventory.

| Agreeableness | Conscientiousness |
|---|---|
| #1: The author can be critical and quarrelsome. | #1: The author is dependable, self-disciplined. |
| #2: The author can be sympathetic and warm. | #2: The author is disorganized, careless. |

| Extraversion | Neuroticism |
|---|---|
| #1: The author is extraverted, enthusiastic. | #1: The author is anxious, easily upset. |
| #2: The author is reserved, quiet. | #2: The author is calm, emotionally stable. |

| Openness | |
|---|---|
| #1: The author is open to new experiences, complex. | |
| #2: The author is conventional, uncreative. | |

Table 8: Items in Ten Item Personality Inventory.

| Agreeableness | Conscientiousness |
|---|---|
| #1: The author sympathizes with others'feelings. | #1: The author often forgets to put things back in their proper place. |
| #2: The author is not interested in other people's problems. | #2: The author is not interested in other people's problems. |
| #3: The author feels others'emotions. | #3: The author makes a mess of things. |
| #4: The author is not really interested in others. | #4: The author likes order. |
| Extraversion | Neuroticism |
| #1: The author is the life of the party. | #1: The author is relaxed most of the time. |
| #2: The author does not talk a lot. | #2: The author seldom feels blue. |
| #3: The author keeps in the background. | #3: The author has frequent mood swings. |
| #4: The author talks to a lot of different people at parties. | #4: The author gets upset easily. |
| Openness | |
| #1: The author has a vivid imagination. | |
| #2: The author is not interested in abstract ideas. | |
| #3: The author has difficulty understanding abstract ideas. | |
| #4: The author does not have a good imagination. | |

Table 9: Items in Mini-PIPI.

Introversion or Extroversion (I/E)

#1: The author is usually: A: "A good mixer with groups of people", B: "Quiet and reserved", or C: "Not sure whether A or B".

#2: Among the authors̀ friends, the author is: A: "Full of news about everybody", B: "One of the last to hear what is going on", or C: "Not sure whether A or B".

#3: The author tends to have: A: "A broad range of friendships with many different people", B: "Deep friendship with very few people", or C: "Not sure whether A or B".

#4: When the author is with a group of people, the author is usually: A: "Join in the talk of the group", B: "Stand back and listen first", or C: "Not sure whether A or B".

#5: The author is: A: "Talk easily to almost anyone", B: "Find a lot to say only to certain people or under certain conditions", or C: "Not sure whether A or B".

#6: In a large group, the author is more often: A: "Introduce others", B: "Get introduced", or C: "Not sure whether A or B".

#7: When the author meets the new people, the author tells what they are interested in: A: "Right away", B: "Only after people to get to know the author", or C: "Not sure whether A or B".

#8: The author is usually: A: "Show their feelings freely", B: "Keep their feelings to themselves", or C: "Not sure whether A or B".

#9: Which word is more suitable for the author: A: "QUIET", B: "HEARTY", or C: "Not sure whether A or B".

#10: Which word is more suitable for the author: A: "RESERVED", B: "TALAKATIVE", or C: "Not sure whether A or B".

#11: Which word is more suitable for the author: A: "CALM", B: "LIVELY", or C: "Not sure whether A or B".

Sensing or iNtuition (S/N)

#1: If the author was a teacher, would they rather teach: A: "Facts-based courses", B: "Courses involving opinion or theory", or C: "Not sure whether A or B".

#2: In doing something that many other people do would the author rather: A: "Invent a way of their own", B: "Do it in the accepted way", or C: "Not sure whether A or B".

#3: Does the author admire more the people who are: A: "Normal-acting to never make themselves the center of attention", B: "Too original and individual to care whether they are the center of attention or not", or C: "Not sure whether A or B".

#4: Does the author usually get along better with: A: "Realistic people", B: "Imaginative people", or C: "Not sure whether A or B".

#5: In reading for pleasure, does the author: A: "Enjoy odd or original ways of saying things", B: "Like writers to say exactly what they mean", or C: "Not sure whether A or B".

#6: Would the author rather be considered: A: "A practical person", B: "An out-of-the-box-thinking person", or C: "Not sure whether A or B".

#7: Would the author rather has a friend: A: "Someone who is always coming up with new ideas", B: "Someone who has both feet on the ground", or C: "Not sure whether A or B".

#8: Which word is more suitable for the author: A: "FACTS", B: "IDEAS", or C: "Not sure whether A or B".

#9: Which word is more suitable for the author: A: "IMAGINATIVE", B: "MATTER-OF-FACT", or C: "Not sure whether A or B".

#10: Which word is more suitable for the author: A: "STATEMENT", B: "CONCEPT", or C: "Not sure whether A or B".

#11: Which word is more suitable for the author: A: "CREATE", B: "MAKE", or C: "Not sure whether A or B".

#12: Which word is more suitable for the author: A: "CERTAINTY", B: "THEORY", or C: "Not sure whether A or B".

#13: Which word is more suitable for the author: A: "FASCINATING", B: "SENSIBLE", or C: "Not sure whether A or B".

#14: Which word is more suitable for the author: A: "LITERAL", B: "FIGURATIVE", or C: "Not sure whether A or B".

Thinking or Feeling (T/F)

#1: Does the author more often let: A: "Their heart rule their head", B: "Their head rule their heart", or C: "Not sure whether A or B".

#2: For the author, which is a higher compliment: A: "A person of real feeling", B: "A consistently reasonable person", or C: "Not sure whether A or B".

#3: Does the author usually: A: "Value emotion more than logic", B: "Value logic more than feelings", or C: "Not sure whether A or B".

#4: Which word is more suitable for the author: A: "CONVINCING", B: "TOUCHING", or C: "Not sure whether A or B".

#5: Which word is more suitable for the author: A: "BENEFITS", B: "BLESSINGS", or C: "Not sure whether A or B".

#6: Which word is more suitable for the author: A: "PEACEMAKER", B: "JUDGE", or C: "Not sure whether A or B".

#7: Which word is more suitable for the author: A: "ANALYZE", B: "SYMPATHIZE", or C: "Not sure whether A or B".

#8: Which word is more suitable for the author: A: "DETERMINED", B: "DEVOTED", or C: "Not sure whether A or B".

#9: Which word is more suitable for the author: A: "GENTLE", B: "FIRM", or C: "Not sure whether A or B".

#10: Which word is more suitable for the author: A: "JUSTICE", B: "MERCY", or C: "Not sure whether A or B".

#11: Which word is more suitable for the author: A: "FIRM-MINDED", B: "WARM-HEARTED", or C: "Not sure whether A or B".

#12: Which word is more suitable for the author: A: "FEELING", B: "THINKING", or C: "Not sure whether A or B".

#13: Which word is more suitable for the author: A: "ANTICIPATION", B: "COMPASSION", or C: "Not sure whether A or B".

#14: Which word is more suitable for the author: A: "HARD", B: "SOFT", or C: "Not sure whether A or B".

Perception or Judging (P/J)

#1: When it is settled well in advance that the author will do a certain thing at a certain time, does the author find it: A: "Nice to be able to plan accordingly", B: "A little unpleasant to be tied down", or C: "Not sure whether A or B".

#2: When the author goes somewhere, would the author rather: A: "Plan what they will do and When", B: "Just go", or C: "Not sure whether A or B".

#3: Does the idea of making a list of what the author should get done over a weekend: A: "Help the author", B: "Stress the author", C: "Positively depress the author", or D: "Not sure whether A, B, or C".

#4: When the author have a special job to do, does the author like to: A: "Organize it carefully before they start", B: "Find out what is necessary as they go along", or C: "Not sure whether A or B".

#5: Does the author prefer to: A: "Arrange picnics, parties etc, well in advance", B: "Be free to do whatever to looks like fun when the time comes", or C: "Not sure whether A or B".

#6: Does following a schedule: A: "Appeal to the author", B: "Cramp the author", or C: "Not sure whether A or B".

#7: Is the author more successful: A: "At following a carefully worked out plan", B: "At dealing with the unexpected and seeing quickly what should be done", or C: "Not sure whether A or B".

#8: In author's daily work, does the author: A: "Usually plan their work so the author won't need to work under pressure", B: "Rather enjoy an emergency that makes their work against time", or C: "Hate to work under pressure", or D: "Not sure whether A, B, or C".

#9: Which word is more suitable for the author: A: "SCHEDULED", B: "UNPLANNED", or C: "Not sure whether A or B".

#10: Which word is more suitable for the author: A: "SYSTEMATIC", B: "SPONTANEOUS", or C: "Not sure whether A or B".

#11: Which word is more suitable for the author: A: "SYSTEMATIC", B: "CASUAL", or C: "Not sure whether A or B".

Table 10: Items in Myers-Briggs Type Indicator.