# OpenReview forum: "PsyCoT: Psychological Questionnaire as Powerful Chain-of-Thought for Personality Detection"
_EMNLP/2023/Conference — EMNLP 2023 Findings_

### Official Review · Reviewer_3hiz · 2023-07-24

**Soundness:** 3

**Excitement:**

4: Strong: This paper deepens the understanding of some phenomenon or lowers the barriers to an existing research direction.

**Paper Topic And Main Contributions:**

The paper describes and evaluates an *NLP engineering experiment*. The authors try to predict the personality  of an author based on text written by the author. Their approach is to use a Large Language Model in an instructive way to rate an authors personality in a multi-turn dialog (following the chain-of-thoughts approach). The LLM, here GPT3.5, is gets first the text in question, some instructions on the answering format and finally multiple request about the authors personality. The rating request are based on psychological questionnaires (i.e., MBTI and Big 5). Finally, the LLM is asked to rate the text authors's personality on different dimensions.

From my point of view, the paper provides two main contributions. First, it evaluates the usage LLMs in order to predict the personality of a person on the basis of text written by that person. Second, it compares two approaches of prompt engineering, one using single-turn instructions and another one using multi-turn instructions.

Finally, both approaches as well as several other are compared in a formative evaluation.


**Questions For The Authors:**

Question A: Why do you not give the results for TrigNet and DDGCN in Table 1 for the Essays dataset, as they work best on the Kaggle dataset?

**Reasons To Accept:**

- The paper describes in a reproducible way an interesting approach to predict aspect of an author's personality from text they have written.
- The approach is validated in a feasible evaluation.
- An ablation study shows the effectiveness go the multi-turn approach.
- The work not only shows the technical possibility to predict personality (i.e., traits) using LLMs, but also the (relative) simplicity, which is in turn a warning for potential misuse.

**Reasons To Reject:**

- In the results section the best two (alternative!) approaches which works best for Kaggle are not show for Essays. Thus the results are not complete.
- A specific discussion is missing and the limitations are very short. Obvious limitations like application to only one language (which is fine, but should be named) or the rather artificial datasets are not mentioned.
- The description of the two used data sets is very short. I would aspect at least additional information on the mean length of the texts to get a feeling for the needed amount of data per text author.

**Reproducibility:**

5: Could easily reproduce the results.

**Reviewer Confidence:**

5: Positive that my evaluation is correct. I read the paper very carefully and I am very familiar with related work.

**Typos Grammar Style And Presentation Improvements:**

- Line 118-121: The sentence sounds somehow incomplete.
- Line 167: the ChatGPT -> remove the
- Line 356: Implementation

---

> ### Author Rebuttal · Authors · 2023-08-28
>
> We sincerely appreciate your time and dedication in reviewing our paper. We are grateful for your recognition of the contributions made by this work. In the following, we carefully address each of your concerns point by point.
>
> #### **Q1: Regarding the results of TrigNet and DDGCN not applied in Essays datasets.**
>
> **A1:** Thank you for the insightful question. We’d like to provide some clarification on this. TrigNet and DDGCN are both designed to amalgamate multiple posts. TrigNet accomplishes this by leveraging a psycholinguistic dictionary (LIWC) to establish a triplet graph between posts, whereas DDGCN autonomously learns to construct a graph among multiple posts. However, it's worth noting that the Essays dataset comprises only a single paragraph from the user, without multiple posts. Consequently, TrigNet and DDGCN are not directly applicable to this dataset.
>
> #### **Q2: Regarding discussion of limitations.**
>
> **A2:** We agree with the reviewer that these two limitations should be mentioned. We will supplement these discussions of limitations in the revised paper.
>
> #### **Q3: Regarding the details of the two data sets.**
>
> **A3:** Thank you for the excellent suggestion. For the Essay dataset, the average token count per sample is 762.38, whereas for Kaggle, it stands at 1632.38. We will provide additional dataset statistics in the Appendix during revision.
>
> #### **Q4: Regarding the typos and improvements.**
>  **A4:** We will revise them accordingly. Thank you for your careful review!
>
> Thanks again for your review! Please let us know if you have any further questions, and we are happy to discuss further.

---

### Official Review · Reviewer_6BCv · 2023-08-03

**Typos Grammar Style And Presentation Improvements:** Line 256
**Soundness:** 3

**Excitement:**

3: Ambivalent: It has merits (e.g., it reports state-of-the-art results, the idea is nice), but there are key weaknesses (e.g., it describes incremental work), and it can significantly benefit from another round of revision. However, I won't object to accepting it if my co-reviewers champion it.

**Missing References:**

The following papers are critically missing:
Automatic Personality Assessment Through Social Media Language - Park et al
VERY similar baseline - Systematic Evaluation of GPT-3 for Zero-Shot Personality Estimation - Ganesan et al

**Paper Topic And Main Contributions:**

This paper proposes a method to use item-based psychology questionnaires as a way to improve personality detection capabilities of LLMs in a CoT framework. They prompt the model to rate individual items from the questionnaire at each turn and leverage the historical rating results to determine the underlying personality trait.

**Questions For The Authors:**

A: The Big 5 personality traits elicit outcomes in a dimensional form. When this method elicits the outcome in the most dimensional form, why do you reduce it back to 2 classes?

B: Post order matters (missing ref: Nelson Liu et al., 2023) matters for these LLMs, given that some messages have more information than others for Personality estimation (missing ref: Lynn et al 2020). How do you account for bias introduced due to order of posts in your prompting setup?

C: Missing row in results table: Item informed high/low classification score (What is the performance when you aggregate the scores for each item)

D: 10 vs 44 item scale is expected to have a big drop. Performance of 20 items scale would be more interesting to see since the 20 item scale has shown convergent validity (ref: Table 1 in Park et al., 2015)

**Reasons To Accept:**

This paper proposes the first use of item-based personality related questionnaires to establish personality traits exhibited in text. Their method works better than simple standard prompting.

**Reasons To Reject:**

CoT does not just guide LLMs through arbitrary reasoning steps, but through a definitive order of reasoning steps. Furthermore, the abstract nature of the personality detection task render the step-by-step nature of the chain-of-thought problem solving approach challenging even for humans.

How can high and low be defined for the Big 5 traits? In its original form, this is a regression task, not a classification one.

There are no statistical significance tests b/w PsyCot and Standard baseline, and PsyCoT and RoBERTa.

The best model for these tasks is a simple regression model - Park et al (mentioned in missing references). However, the paper does not compare to that model and neither does it compare to Lynn et al 2020, which is another strong performing model. It is hard to contextualize the results in this paper and pit them against what the community knows.

**Reproducibility:**

4: Could mostly reproduce the results, but there may be some variation because of sample variance or minor variations in their interpretation of the protocol or method.

**Reviewer Confidence:**

4: Quite sure. I tried to check the important points carefully. It's unlikely, though conceivable, that I missed something that should affect my ratings.

---

> ### Author Rebuttal · Authors · 2023-08-28
>
> Thanks for your perspective comments. We carefully address each of your concerns point by point as follows.
>
> #### **Q1: Regarding the CoT in personality detection.**
> **A1:** Thank you for the insightful perspective. We acknowledge the inherent difficulty in developing a chain-of-thought solution for personality detection. This is precisely why we have meticulously crafted the PsyCoT approach. It assesses individual personality traits by incorporating a series of psychological questionnaire items into the prompt, effectively harnessing the capabilities of Large Language Models (LLMs) for this task. These items can be viewed as a well-structured collection forming a chain-of-thought (CoT).
>
> #### **Q2: Regarding the form of Big 5 traits.**
> **A2:** Although the original tasks of the Big 5 are in the regression form, this data is further collected and processed into the binary label format (Y/N), without releasing the original scores from the raters. Consequently, we have framed this task as a classification problem, designating 'High' as 'Y' and 'Low' as 'N.' While it is still possible to approach it as a regression task by assigning 'Y' as 1 and 'N' as 0, previous research, such as the work by Ganesan et al. [1], has delved into the distinctions between classification and regression in similar contexts, ultimately suggesting that simplifying the task to classification is more desirable. We appreciate your question and will provide further clarification in the revision.
>
> [1] Ganesan A V, Lal Y K, Nilsson A H, et al. Systematic Evaluation of GPT-3 for Zero-Shot Personality Estimation[J]. arXiv preprint arXiv:2306.01183, 2023.
>
> #### **Q3: Regarding the statistical tests.**
> **A3:** Thank you for the valuable suggestion. Regarding PsyCoT and other in-context learning-based baselines, we have indeed set the temperature to 0 to ensure deterministic outputs when interacting with the API, which can be subject to minor randomness. Additionally, we've executed multiple runs of the baselines, including RoBERTa, and consistently observed improved performance with our approach compared to the baselines. To enhance the rigor of our analysis, we will incorporate the results of statistical testing in the revised version.
>
> #### **Q4: Regarding the baselines.**
> **A4:** Thank you for bringing up these two methods. We’d like to provide some clarification here. Firstly, the model proposed by Park et al. (2014) is better suited for regression tasks and is therefore not well-matched to the datasets used in this study, which involve classification labels (as previously mentioned in A2). Secondly, regarding the method presented by Lynn et al. (2020), previous research, as demonstrated by the results of TrigNet (2021) and DDGCN (2023), has consistently shown superior performance when compared to Lynn et al. (2020) (detailed results on Kaggle are provided below). Consequently, we have omitted Lynn et al. (2020) from our selection of baselines in favor of these more stronger methods. To ensure completeness, we will include these additional results in the revised version.
>
> |  Methods | I/E | S/N | T/F | J/P | Average |
> | ---| :-: | :-: | :-: | :-: | :-: |
> | **Lynn et al.** | 65.43 | 62.15 | 78.05 | 63.92 | 67.39 |
> | TrigNet | 69.54 | 67.17 | 79.06 | 67.69 | 70.86 |
> | DDGCN | 69.52 | 67.19 | 80.53 | 68.16 | 71.35 |
>
> #### **Q5: Regarding the form of this task.**
> **A5:** Our study indicates that simplifying multi-label personality detection into multiple binary classification tasks leads to performance improvements (the average results on standard prompt are reported below). Hence, this approach was chosen for our experiments. We will supplement these additional results and provide more discussion in the revision.
>
> |  Forms  | Accuracy | Macro-F1 |
> | --- | :-: | :-: |
> | **Multi-label** | 51.74 | 46.85 |
> | Five binary (ours) | 58.70 | 54.20 |
>
> #### **Q6: Regarding the post order.**
> **A6:** This is an insightful question! We fully concur with the reviewer that post order can affect the model performance. In Section 5.3, we have conducted a preliminary investigation into the influence of post order, and our experimental results highlight that in-context learning-based methods can be affected by it. Nevertheless, we would like to acknowledge that delving into the determination of the optimal post order falls outside the scope of our current study. This aspect stands as one of our future research directions.
>
> #### **Q7: Regarding the aggregation of item scores.**
>  **A7:** Thank you for the excellent suggestion. In fact, we have computed the aggregated trait scores and visualized their distributions in Figure 3. Our analysis reveals that the aggregated scores for Agreeableness, Conscientiousness, and Neuroticism cross the dashed line (3=Neutral) and exhibit a shift towards one side. This observation suggests potential bias in the LLM's ratings of the questionnaires. As a result, it becomes imperative to determine a suitable threshold for converting these scores into binary labels. We appreciate your suggestion, and in our revision, we will also incorporate the aggregate scores into the tables.
>
> #### **Q8: Regarding the 20 item scale.**
> **A8:** Thank you for the great suggestion. We evaluate the performance of PsyCoT on the Essays dataset using a 20-item scale from Mini-IPIP [2]. As shown in below table, the results are better than the 10-item scale, but still inferior to the 44-item scale. We will supplement the results in the revised version as suggested.
>
> |  Item-scale  | Accuracy | Macro-F1 |
> | :-: | :-: | :-: |
> | 10 | 58.46 | 54.72 |
> | **20** | 59.19 | 57.42 |
> | 44 | 59.64 | 58.43 |
>
> [2] Donnellan M B, Oswald F L, Baird B M, et al. The mini-IPIP scales: tiny-yet-effective measures of the Big Five factors of personality[J]. Psychological assessment, 2006, 18(2): 192.
>
> #### **Q9: Regarding the missing references.**
> **A9:** We will include these references in the revised paper accordingly. Thank you!
>
> We sincerely thanks the reviewer again for your valuable suggestions. Please let us know if you have any further questions, and we are happy to discuss further.

---

### Official Review · Reviewer_QAsf · 2023-08-12

**Soundness:** 3

**Excitement:**

4: Strong: This paper deepens the understanding of some phenomenon or lowers the barriers to an existing research direction.

**Paper Topic And Main Contributions:**

The paper proposes PsyCoT, a chain-of-thought method specialized in personality detection tasks. The methodology uses chain-of-thought reasoning in the form of answering psychological questionnaires. The authors investigate the correlation
between the trait score and the chosen personality and explore the patterns of score distributions in terms of different traits.


**Reasons To Accept:**

The methodology is interesting enough to understand the ability of LLMs for personality detection, and the explanations about details are clear. The authors analyze the correlation between scores and traits to predict some patterns between the personality traits and the LLM's behaviors. Overall, it is easy to follow and good enough to inspire researchers in a similar field to analyze or predict authors' personalities.

**Reasons To Reject:**

PsyCoT is basically coming from a popular method, Chain-of-Thought. It is weak to say the proposed method is innovative and original though the application to personality detection has been yet underexplored.

**Reproducibility:**

4: Could mostly reproduce the results, but there may be some variation because of sample variance or minor variations in their interpretation of the protocol or method.

**Reviewer Confidence:**

3: Pretty sure, but there's a chance I missed something. Although I have a good feel for this area in general, I did not carefully check the paper's details, e.g., the math, experimental design, or novelty.

---

> ### Author Rebuttal · Authors · 2023-08-28
>
> We thank you for your time and effort in reviewing our paper. We are grateful that you find our work to be both interesting and inspiring.
> #### **Q1: Regarding the innovation of this work.**
> **A1:** We appreciate the valuable feedback. While we acknowledge that our work builds upon Chain-of-Thought, we would like to emphasize the distinct contributions and unique value our research brings to the community.
>
> **Limitation and Challenge of Existing Work** Prior studies in personality detection have typically relied on crafting specialized models using datasets sourced from the internet. The limitations of these datasets, both in terms of their quality and their alignment with personality traits, have significantly restricted the attainment of optimal results.
>
> **Our Contribution** We introduce a pioneer approach that leverages the capabilities of Large Language Models to effectively analyze and elucidate personality traits. Our PsyCoT model emulates the cognitive process of human self-assessment related to personality traits. This emulation is accomplished through a dynamic, multi-turn dialogue process, which rigorously employs reasoning steps using items from a psychological questionnaire. This approach results in a significantly enhanced alignment with personality traits.
>
> **Impact of our Work** Our experimental results demonstrate the effectiveness and superiority of our approach over state-of-the-art methods. Additionally, PsyCoT exhibits versatility beyond personality detection, with applications extending to questionnaire assessment and depression assessments. Finally, we believe our research offers valuable insights to the personality detection community, serving as a catalyst for future explorations in this domain.
>
> Thanks again for your review! Please let us know if you have any further questions, and we are happy to discuss further.

---

### Meta-Review · Area_Chair_52Zu · 2023-09-18

**Recommendation:** 4

**Metareview:**

This paper applies Chain of Thought reasoning to personality detection using large language models. Their method is based on commonly used personality questionnaires (MBTI and Big 5), but re-structured for use with  LLMs in a multi-turn dialogue. The reviewers initially raised issues related to comparisons to previous work, which were addressed by subsequent experiments during the discussion period. The reviewers also questioned the framing of the task as Chain of Thought, given the lack of a concrete reasoning process. This point was discussed at length, and the authors promised to reconsider their framing. Overall, reviews are positive.

---

### Decision · Program_Chairs · 2023-10-07

**Decision:**

Accept-Findings

**Comment:**

This paper applies Chain of Thought reasoning to personality detection using large language models. Their method is based on commonly used personality questionnaires (MBTI and Big 5), but re-structured for use with  LLMs in a multi-turn dialogue. The reviewers initially raised issues related to comparisons to previous work, which were addressed by subsequent experiments during the discussion period. The reviewers also questioned the framing of the task as Chain of Thought, given the lack of a concrete reasoning process. This point was discussed at length, and the authors promised to reconsider their framing. Overall, reviews are positive.